# High-Throughput Fingerprinting of Rhizobial Free Fatty Acids by Chemical Thin-Film Deposition and Matrix-Assisted Laser Desorption/Ionization Mass Spectrometry

**DOI:** 10.3390/mps3020036

**Published:** 2020-05-04

**Authors:** Aleksey Gladchuk, Julia Shumilina, Alena Kusnetsova, Ksenia Bureiko, Susan Billig, Alexander Tsarev, Irina Alexandrova, Larisa Leonova, Vladimir A. Zhukov, Igor A. Tikhonovich, Claudia Birkemeyer, Ekaterina Podolskaya, Andrej Frolov

**Affiliations:** 1Institute of Toxicology, Federal Medical-Biological Agency of Russia, 192019 Saint Petersburg, Russia; aleglad24@gmail.com (A.G.); analekt@mail.ru (I.A.); ek.podolskaya@gmail.com (E.P.); 2Department of Biochemistry, St. Petersburg State University, 199034 Saint Petersburg, Russia; schumilina.u@yandex.ru (J.S.); alena_kyy@mail.ru (A.K.); ksenya.bu@gmail.com (K.B.); alexandretsarev@gmail.com (A.T.); larisa_leonova@mail.ru (L.L.); 3Department of Bioorganic Chemistry, Leibniz Institute of Plant Biochemistry, 06120 Halle, Germany; 4Institute of Analytical Chemistry, Faculty of Chemistry and Mineralogy, Universität Leipzig, 04103 Leipzig, Germany; billig@uni-leipzig.de (S.B.); birkemeyer@chemie.uni-leipzig.de (C.B.); 5All-Russia Research Institute for Agricultural Microbiology, 196608 Saint Petersburg, Russia; vladimir.zhukoff@gmail.com (V.A.Z.); arriam2008@yandex.ru (I.A.T.); 6Department of Genetics and Biotechnology, St. Petersburg State University, 199034 Saint Petersburg, Russia; 7Institute of Analytical Instrumentation, Russian Academy of Sciences, 198095 Saint Petersburg, Russia

**Keywords:** bacteria, barium monocarboxylates, chemical deposition technique, free fatty acids (FFAs), Langmuir film technology, matrix-assisted laser desorption/ionization-time of flight mass spectrometry (MALDI-TOF-MS), metabolic fingerprinting, rhizobia

## Abstract

Fatty acids (FAs) represent an important class of metabolites, impacting on membrane building blocks and signaling compounds in cellular regulatory networks. In nature, prokaryotes are characterized with the most impressing FA structural diversity and the highest relative content of free fatty acids (FFAs). In this context, nitrogen-fixing bacteria (order Rhizobiales), the symbionts of legumes, are particularly interesting. Indeed, the FA profiles influence the structure of rhizobial nodulation factors, required for successful infection of plant root. Although FA patterns can be assessed by gas chromatography—(GC-) and liquid chromatography—mass spectrometry (LC-MS), sample preparation for these methods is time-consuming and quantification suffers from compromised sensitivity, low stability of derivatives and artifacts. In contrast, matrix-assisted laser desorption/ionization-time of flight mass spectrometry (MALDI-TOF-MS) represents an excellent platform for high-efficient metabolite fingerprinting, also applicable to FFAs. Therefore, here we propose a simple and straightforward protocol for high-throughput relative quantification of FFAs in rhizobia by combination of Langmuir technology and MALDI-TOF-MS featuring a high sensitivity, accuracy and precision of quantification. We describe a *step-by-step* procedure comprising rhizobia culturing, pre-cleaning, extraction, sample preparation, mass spectrometric analysis, data processing and post-processing. As a case study, a comparison of the FFA metabolomes of two rhizobia species—*Rhizobium leguminosarum* and *Sinorhizobium meliloti*, demonstrates the analytical potential of the protocol.

## 1. Introduction

Fatty acids (FAs) represent one of the most important metabolite classes in living organisms [1]. Indeed, on one hand, these compounds are readily involved in a wide array of enzymatic reactions yielding esters of aliphatic, cyclic and aromatic alcohols determining the structure of membrane and storage lipids [2]. On the other, in the non-esterified form, FAs serve as precursors of hormones and modulators—key players of plant [3] and animal [4] regulatory pathways, involved in all vital physiological reactions. Not less important is the role of free fatty acids (FFAs) as low molecular weight effectors, directly involved in immune responses across all kingdoms of living organisms [4]. Therefore, FFAs attract the special attention of many biologists and analysts [5].

The cellular pool of fatty acids can be relatively easily accessed either directly (FFAs) or after degradation of esters (bound, esterified FA fraction) [6]. Due to a high structural heterogeneity of the ester-bound fraction and essential contribution of FFAs to the overall FA patterns [7], prokaryotes represent a remarkable group of organisms with respect to their FA composition. Bacterial lipids are composed of phospholipids, their aminoacylated derivatives, glycerolipids, betaine lipid diacylglyceryl-*N,N,N*-trimethylhomoserine (DGTS), as well as ornithine lipids, lipopolysaccharides, sphingolipids, sulfonolipids and phenolic lipids [8].The patterns of bacterial FFAs are strongly dominated by higher, linear (saturated C12–C28 and unsaturated C14–C28), branched (mostly saturated, containing up to 52 carbon atoms), hydroxylated and cyclic acids [7]. Regarding their FA patterns, rhizobial bacteria represent one of the most interesting groups of prokaryotes (Figure 1). These microorganisms readily form symbiotic associations with roots of higher plants, most often legume species [9]. Formation of root nodules, i.e., morphological structures, characteristic for legume rhizobial symbiosis, is a complex process, at the early steps governed by so-called nodulation (Nod) factors—sulfated chitooligosaccharides which are, for example, in the case of *Rhizobium meliloti*, mono-*N*-acylated by unsaturated C16 or by a series of C18 to C26 (ω-1)-hydroxylated fatty acids [10]. Therefore, analysis of FA composition in rhizobial cells remains an important task of analytical science.

In modern bioanalytical chemistry, analysis of FFA composition most often relies on gas chromatography—mass spectrometry (GC-MS) [11]. While short-chain volatile acids can be analyzed by headspace injection techniques [12], long-chain ones can be assessed by liquid injection after appropriate derivatization [13]. The latter approach, relying on detection of methyl or trimethylsilyl esters, proved to be efficient in analysis of branched fatty acids in bacterial membranes [14]. However, implementation of derivatization procedures in experimental workflows dramatically reduces sample throughput and might trigger transesterification of lipids, as well as isomerization and oxidation of unsaturated FFAs [15,16]. Although hydrophilic-interaction [17] and reversed phase liquid chromatography (HILIC and RP-LC, respectively) [18], coupled on-line to electrospray ionization mass spectrometry (ESI-MS) or tandem MS (MS/MS) in a multiple reaction monitoring (MRM) mode [19] can be also employed in quantification of FFAs, these techniques lack sensitivity or fatty acid metabolome coverage in comparison to traditional GC-MS-based workflows [20].

These bottlenecks of GC-MS- and LC-MS-based approaches for FA analysis can be efficiently overcome by implementation of matrix-assisted laser desorption/ionization-time of flight mass spectrometry (MALDI-TOF-MS) in combination with Langmuir–Blodgett technology [21]. Recently, we have shown that Langmuir–Blodgett films can be not only efficiently implemented in selective enrichment of phosphopeptides [22], protein adducts of organophosphorous compounds [23] and chlorinated insecticides [24], but also proved to be an efficient tool for high-throughput and sensitive fingerprinting of free fatty acids as their barium monocarboxylates in positive ion mode [25]. Therefore, here we implement this approach in a newly developed protocol for the analysis of FFA composition in rhizobial bacteria and propose a comprehensive quantitative solution for fingerprinting of FFAs in cultured cells by MALDI-TOF-MS. Thereby, we demonstrate for the first time, the applicability of the Langmuir/MALDI-TOF-MS approach to answering real biological questions.

## 2. Experimental Design

Langmuir technology in combination with MALDI-TOF-MS is the core methodology, underlying the protocol, presented here [25]. In general, it gives access to unique structures, characterized with a high regularity at the molecular level [26]. This can be exemplified by formation of monolayers, accompanying the reaction of stearic acid with a trivalent metal salt (e.g., lanthanum trinitrate) at the interface of organic and aqueous phases [23]. Such monolayers are well-accessible under standard laboratory conditions. The preparation protocol typically assumes an overlay of an aqueous salt buffer with FA in hexane or other organic solvent [27]. Under these conditions, the hydrophilic carboxylic groups are involved in hydrogen bonding with water and builds ion pairs with dissolved multivalent cations, whereas the non-polar aliphatic chain interacts with the organic phase. Formation of these ion pairs is fast, quantitative and yields non-soluble in water (but soluble in organic solvents) monolayers for long-chain FAs (C12 and higher). Importantly, only free fatty acids are involved in this reaction, whereas most of the other metabolites are not incorporated in the monolayers. After mechanical collapsing, such monolayers can be collected in polypropylene tubes, reconstituted and analyzed by MALDI-TOF-MS. 

Although the soft reaction conditions make Langmuir technology a promising tool in sample preparation for mass spectrometric analysis, its implementation in analytical practice required, however, principal changes in the monolayer preparation strategy. First, as we have shown in our recent work [21], the cationic component needs to be replaced by a divalent metal, ideally barium. Indeed, these cations form mostly monocarboxylates and no mixed salts, i.e., yielding efficiently ionizable monolayers. Such monocarboxylate monolayers can be desorbed from the surface of a conventional MALDI target by UV laser, and can be, therefore, be analyzed by MALDI-TOF-MS instrumentation. Ionization efficiency can be further increased by supplementation of the aqueous phase with conventional MALDI matrices, based on substituted benzoic or cinnamic acids, which improve energy distribution in the sample upon the laser shot [28]. In this context, 2,5-dihydroxybenzoic acid (DHB) proved to be the best choice for FAs due to its good solubility in water [25]. The resulting spectra are well interpretable and can be used for quantitative estimations.

Unfortunately, short-chain and unsaturated FAs do not form solid monolayers; their salts remain on the surface of the aqueous phase, when such monolayers are collapsed. This and several other limitations can be overcome by transfer of the Langmuir technology from the planar setup (e.g., in a Petri dish) to a droplet format [25]. This allows formation of monolayers directly on a MALDI target without any losses of the sample. Secondly, due to the convex shape of the droplet surface, higher concentrations of the metal ions at the interphase can be achieved. Thereby, sample consumption is rather low (not more than 1 µL per sample). Finally, the structure of the monolayers remains regular, resulting in a high reproducibility of analysis. In additional experiments, we addressed the limitations of the method in the context of the acid chain length. We found, that the C6:0 and C8:0 acids are not detectable by this protocol, whereas the signal of C10:0 demonstrates lower intensity in comparison to higher homologs (C12:0 and higher).

The overall experimental setup behind the proposed protocol includes several principal steps (Figure 2): culturing rhizobia on agar and in aqueous nutritional medium, pre-cleaning the bacterial cells, hexane extraction of the aqueous suspension from the bacterial pellet, application of the sample to the MALDI target, mass spectrometric analysis, qualitative and quantitative analysis and statistical interpretation (post-processing).

### 2.1. Materials and Chemicals


Safe-lock 2 mL polypropylene tubes (Eppendorf, obtained from Helicon, Moscow, Russia; Cat. No.: Epp 0030 120.094);Cell culture dishes, 90 × 14 mm (Noex Pharma, Moscow, Russia);n-Hexane (HPLC grade, ≥95% purity, Merck, obtained from ChimMed, St. Petersburg, Russia; Cat. No.: 270504);Dipotassium hydrogen phosphate anhydrous (≥98%, ACS grade, VWR International, obtained from Helicon, Moscow, Russia; Cat. No.: Am-O705-0.5);Agar for bacteriological works, solid (VWR International, obtained from Helicon, Moscow, Russia; Cat. No.: Am-J637-0.5);Magnesium sulfate anhydrous (reagent grade, ≥97%, Merck, obtained from ChimMed, St. Petersburg, Russia; Cat. No.: 208094);Calcium carbonate (ACS reagent grade, ≥99.0%, Merck, obtained from ChimMed, St. Petersburg, Russia; Cat. No.: 239216);Sodium chloride (ACS reagent grade, ≥99.0%, Merck, obtained from ChimMed, St. Petersburg, Russia; Cat. No.: S9888);D-Mannitol (≥98%, Merck, obtained from ChimMed, St. Petersburg, Russia; Cat. No.: M4125);Microgranulated yeast extract (Biospringer, acquired from Helicon, Russia; Cat. No.: H-0601MG-0.5);Barium acetate (Merck, obtained from ChimMed, St. Petersburg, Russia; Cat. No.: 243671);Parafilm M (Bemis Company, obtained from Helicon, Moscow, Russia; Cat. No.: PM996);2,5-Dihydroxybenzoic acid (mass spectrometry grade, Bruker Daltonik GmbH, Bremen, Germany; Cat. No.: 8201346);Acetonitrile (LC-MS grade, Merck, obtained from ChimMed, St. Petersburg, Russia; Cat. No.: 100029);Bacterial strains Rhizobium leguminosarum bv. viciae RCAM1026 and Sinorhizobium meliloti RCAM1021 (collection of the All-Russian Research Institute for Agricultural Microbiology, St. Petersburg, Russia) kept at 4 °C in Petri dishes on agar nutrient medium.


### 2.2. Equipment


Vortex mixer Vortex-genie 2 (Scientific Industries, NY, USA);Ultrasound bath Sonorex TK-30 (BANDELIN electronic GmbH & Co. KG, Berlin, Germany);Benchtop incubator shaker Innova 40 (Eppendorf, obtained from Helicon, Moscow, Russia);Tabletop centrifuge (5415 R, Eppendorf, obtained from Helicon, Moscow, Russia);MTP 384 polished steel BC target (Bruker Daltonik GmbH, Bremen, Germany);MALDI TOF/TOF mass spectrometer UltrafleXtreme (Bruker Daltonik GmbH, Bremen, Germany);Microplate spectrophotometer PowerWave HT (BioTek Instruments, obtained from Bioline LLC, St. Petersburg, Russia).


### 2.3. Software


FlexControl 3.4 and FlexAnalysis 3.4 (Bruker Daltonik GmbH, Bremen, Germany);Progenesis MALDI v1.2 (Nonlinear Dynamics, Newcastle upon Tyne, UK);MetaboAnalyst 4.0 (https://www.metaboanalyst.ca/MetaboAnalyst/home.xhtml).


### 2.4. Costs

The overall price of consumables for analysis of one sample can be estimated as 2 €. These costs include reagents and plastic ware (tubes, pipette tips).

## 3. Procedure

### 3.1. Sterilize (180 min)

Prepare and sterilize nutritional medium # 79 (0.003 mol/L K_2_HPO_4_, 0.002 mol/L MgSO_4_ x 7H_2_O salts, 0.002 mol/L NaCl, 0.001 mol/L CaCO_3_, 0.055 mol/L mannitol, 0.4 g/L yeast extract), tips and tubes by autoclaving at 121 °C during 20 min.

### 3.2. Prepare Rhizobia Culture in Liquid Nutritional Medium (Two Days)


Take several colonies of *R. leguminosarum* or *S. meliloti* from the surface of agar medium with an inoculation loop or sterilized spatula and transfer them to the liquid nutritional medium # 79 kept at 28 °C.Grow bacteria at 28 °C under continuous shaking (100 rpm) in a glass flask with a breathable cover during 48 h to achieve an optical density (OD620) of 0.25 at 620 nm.


### 3.3. Estimate Titer (As Optical Density, OD) of Rhizobial Culture (60 min)

Pipette 250 µL of each rhizobial culture and nutritional medium as control (n = 4 or more) in the wells of a 96-well microtiter plate.Determine the titer (OD620) of the bacterial culture spectrophotometrically. If OD620 is 0.250 ± 0.005, proceed to the next step. If OD620 is higher or lower, centrifuge the suspension (for 10 min at 4000× *g*/25 °C) and re-suspend the cells in a calculated volume of culturing medium. The resulting bacterial culture sediment is diluted with a nutrient solution, the volume of which is calculated by the formula:*V* = *d* × *V_0_*/*d_a_*,(1)
where *V_0_*—the initial volume of the bacterial culture solution and the nutrient medium, *V*—the volume of the nutrient medium solution added after centrifugation, *d —* is the relative optical density of the initial solution of bacterial culture and the nutrient medium, *d_a_ —* the target relative optical density of the bacterial culture solution and the nutrient medium at which the concentration of bacterial culture will have desired value. After dilution, a repeated measurement of the optical density *d* of the resulting bacterial culture solution is performed. If the measured value of *d* differs from *d_a_*, then the procedure described above is repeated until the condition |*d*−*d_a_*| ≤ *ε_a_* is satisfied, where *ε_a_* is ± 0.005.

### 3.4. Pre-Clean of Bacterial Cells Prior to Extraction (60 min)

All works need to be done under a fume hood as hexane vapors are toxic.Add 0.4 mL *n*-hexane to the 2 mL *safe-lock* polypropylene tubes, cover the tubes with parafilm and vortex (1000 rpm) for 1 min. Discard *n*-hexane and dry the tubes under airflow. This step is necessary to wash out potential hexane-soluble contaminations, present in tubes.Transfer 2 mL rhizobial culture to the tubes, and centrifuge for 10 min at 4000× *g*/25°C.Discard supernatants, re-suspend the pellets in 1 mL nutritional medium # 79, prepared without addition of mannitol and yeast extract (pH 7.0–7.2), vortex (1000 rpm) for 1 min and centrifuge for 10 min at 4000× *g* and 25 °C. Repeat this procedure two times and discard the supernatant after the last centrifugation.Add 1 mL 0.9% (*w/v*) NaCl to the bacterial pellets, vortex (1000 rpm) for 1 min, centrifuge (10 min, 4000× *g*, 25 °C) and discard supernatant. Repeat this procedure twice and discard the supernatant after the last centrifugation.

The first pre-cleaning step is required to remove the components of the culture medium and excreted bacterial metabolites, potentially interfering with the analysis of FFAs. The use of the isotonic medium ensures preserving the osmotic potential of the rhizobial cells and prevents loss of FFAs due to loss of cell integrity.

### 3.5. Extract Fatty Acids from the Bacterial Cells with N-Hexane

Two extraction workflows were employed to access the FFA fraction of rhizobial cells.

#### 3.5.1. Extraction from N-Hexane Lysates (15 min)


Supplement the pre-cleaned bacterial pellets with 0.4 mL *n*-hexane, cover the tubes with parafilm and vortex (1000 rpm) for 1 min.Sonicate the resulting suspension for two min in an ultrasonic bath. Add ice to the bath to avoid overheating of the samples.Vortex (1000 rpm) the samples for further 2 min, centrifuge (10 min, 4000× *g*, 25 °C), and collect the supernatant in new 2 mL *safe-lock* polypropylene tubes.


#### 3.5.2. Extraction from Aqueous Lysates (15 min)


Supplement the pre-cleaned pellets with 0.4 mL deionized water and vortex (1000 rpm) for 1 min.Sonicate the resulting suspension 2 min in an ultrasonic bath. Add ice to the bath to avoid overheating of the samples.Add 0.4 mL *n*-hexane to the suspensions, cover the tubes with parafilm and vortex (1000 rpm) the samples for further 2 min, centrifuge (10 min, 4000× *g*, 25 °C), and collect the upper hexane phase in new 2 mL *safe-lock* polypropylene tubes.


### 3.6. Apply the Samples on a MALDI Target (10 min Per Sample, with Increase of Sample Number Time Requiered for Each One is Less)


Apply 0.6 µL of the aqueous mixture containing 2,5-dihydroxybenzoic acid (DHB) and barium acetate (0.25 g/L each) on a spot of a polished stainless steel 384-well MALDI target. The mixture forms an aqueous drop within the spot. Make at least three individual applications (technical replicates) at three different spots for each biological replicate.Apply 0.6 µL of the sample in *n*-hexane on the top central surface of the aqueous drop, containing DHB and barium acetate. Add another 0.6 µL-portion of the sample after complete evaporation of hexane (assessed visually).Apply 2 µL 90% (*v/v*) aq. acetonitrile after complete drying of the spot surface (both hexane and aqueous layers need to evaporate completely). Repeat this procedure after reconstitution of the dried monolayer and complete evaporation of the solvent. Formation of symmetric round droplets is desired at this step.Using the pipette tip, shift the formed droplet to the edge of the spot and wait for evaporation of the solvent. This will reduce matrix effects due to removal of polar contaminants from the MALDI spot.Load the target in the mass spectrometer.


### 3.7. Acquire Mass Spectral Data (20 min)


Adjust laser fluence to 100%. This value will result in better inter-replicate reproducibility.Accomplish automatic registration of spectra by the AutoXecute tool in FlexControl software (UltrafleXtreme instrument, Bruker Daltonics). For this, set acquisition parameters as follows: number of shots accumulated—35,000 (this number of shots is saturating, i.e., does not affect relative intensities of signals in a broad concentration range of analytes), *m/z* range—360–550, type of movement—random walk (complete sample), limit diameter of acquisition area—2000 µm (all other MS settings are summarized in the Appendix A). Start spectra acquisition.Acquire at least three spectra per spot, verify reproducibility of the analysis.Open acquired spectra in FlexAnalysis software. Export mass spectral data to mzXML format for processing of the data.


### 3.8. Data Processing with MALDI Progenesis Software (30 min)


Download the necessary data in mzXML format to Progenesis MALDI. Set the parameter “bin size” to 1.Perform pre-processing of the data to remove noise and background artifacts from the spectra. Set the parameters “Noise Filter” and “Background Top Hat filter” to 4 and 60, respectively. Press the “Pre-Process All Spectra” button and wait until the process is finished. Press “Section Complete” to move to the next step.Perform alignment of the spectra. Set the parameters “Search Area” and “Iterative Cycles” to 5. Press the “Align Spectra” button and wait until the process is finished. Press “Section Complete” to move to the next step.For peak detection choose “Whole Protein” detection method. Set the “Threshold” to 500; set minimum and maximum *m/z* values to 360 and 550, respectively. Press the “Detect Peaks” button and wait until the process is finished. Press “Section Complete” to move to the next step.Group technical replicates that represent each biological replicate. To create the sample, select the necessary spectra listed in the right window and press the “Add Selected Spectra to a Sample” button. Fill in the appropriate name of the sample by clicking on the name “New Sample 1” in the list on the left. Do the same for the other spectra. Press “Section Complete” to move to the next step.Classify samples into groups. Select the necessary samples from the list in the right window and press the “Add Selected Sample to a Group” button. Click on the name “New Group 1” and fill in the appropriate name of the group. Do the same for the other samples. Press “Section Complete” to move to the next step.Review the results of the analysis. Choose “Normalized Peak Height” as a statistics measurement for assessing the differences. Choose the type of normalization. Click the tick box on the peaks of interest to put them in the report. Copy normalization data for peaks of interest to an Excel file for further data analysis by multivariate statistics (these files can be used for further statistical interpretation—see Section 3.9). The normalization of the data can rely on the total ion current (TIC) and on the peak heights of the signals related to barium monocarboxylates of the signals of three typically most abundant fatty acids in the samples: palmitic acid (*m/z* 393.14), oleic acid (*m/z* 419.16) and stearic acid (*m/z* 421.17). Press “Section Complete” to move to the next step.Press “Section Complete” to skip “Stats” section and move to the next step.Create a report from your experimental results. Print the title of your report in the corresponding window and press the “Create Report” button. Afterwards, you can view and print your report in the “Report Output” tab. In addition, you can save the created report in PDF format by pressing the “Save Report” button.


### 3.9. Characterize Differences between Sample Groups by Multivariate Statistics


Rearrange the MS-Excel data exported from Progenesis MALDI at the previous step to a format compatible with the input requirements of the MetaboAnalyst software tool. For this, create an MS Excel table for each type of normalization. The table needs to contain information on the technical and biological replicates in the first row, sample groups in the second one, and the rest of the rows should contain names of the analytes along with the corresponding normalized peak heights. It is possible to perform analysis using all the replicates separately or using only average peak heights within each technical replicate. For the details of the table arrangement see Appendix A.Individually upload the resulting tables to the MetaboAnalyst on-line platform (in the “Statistical Analysis” module). Select the type of data scaling/normalization to access the desired group homogeneity. The following options can be considered: Pareto scaling, range scaling and generalized logarithm transformation.Accomplish principal component analysis (PCA) and hierarchical clustering, build corresponding graphs—scores plots and heat maps. Review the results and create a report.


## 4. Results and Discussion

To illustrate the potential of our protocol, we performed a comparative study of the FFA metabolomes of two close rhizobial species—*Rhizobium leguminosarum* bv. viciae RCAM1026 and *Sinorhizobium meliloti* RCAM1021. For this, the whole workflow was applied to the cultures of these two organisms (*n* = 4, MALDI-TOF-MS analyses were done in three sample application and three spectrum acquisition replicates, all nine technical replicates were treated equally).

### 4.1. Identification of FFAs in Rhizobial Extracts

The analyses revealed nine fatty acid signals in total, which were annotated in the both species by their exact *m/z* values with a mass accuracy better than 10 ppm (Table 1), i.e., within the instrument specifications. Thereby, the quality and absolute intensities of the acquired spectra (Figure 3) were comparable with those observed earlier with collapsed solid monolayers of saturated fatty acids [21] and real extracts obtained from different biological objects including root nodules of pea (*Pisum sativum* L.) [25]. It is important to note, however, that lysis under aqueous conditions (Figure 3A,B) yielded approximately doubled spectral intensities and better signal to noise ratios in comparison to those obtained after hexane lysis (Figure 3C,D and Appendix A–2). Most likely, it can be explained with a higher efficiency of lipid extraction with *n-*hexane that might compromise crystallization and energy transfer from DHB to the analytes or ion suppression. Comparison of the spectra, acquired from different species under the same conditions (Figure 3A–D), revealed identical qualitative signal patterns, i.e., no FFAs, characteristic for only one of the analyzed species were found. The annotation of most analytes could be confirmed by tandem mass spectrometry in post-source decay (PSD) fragmentation mode (Figure 4). The spectra were informative and allowed for the localization of the double bond in the FA structure, as described in our previous work [25] based on the mechanism assuming double bond migration with subsequent σ-cleavage or coordination of the metal by the double bond [29].Verification of the observed signals with the spectrum of corresponding authentic standards (as exemplified for linoleic acid in Figure 4) clearly indicated the reliability of the MS/MS-based annotation of FFAs in the samples, prepared by means of the Langmuir technology, which obviously, represents a principal advantage of the method. The spectra dominated with intense signals at *m/z* 373.02 and 399.04. The first of them corresponds to the loss of CO_2_, formed by the mechanism, earlier reported by Afonso et al. [30] and Crockett et al. [31]. The second signal might be interpreted as a water loss, underlied by a keto-enol tautomeric transition, which is accompanied by transfer of a hydrogen atom from the α-methylene group to the carboxyl oxygen, with subsequent dehydration and formation of barium oxyacetylenide.

However, some contaminations were introduced in the sample with the DHB matrix, mainly relatively low abundant signals, like *m/z* 325 (Appendix A-7, Appendix A-7). However, the origin of the broad signal at *m/z* 300 could not be understood, although it was present in most of the analyzed MS/MS spectra. One needs to keep in mind, however, that the LIFT system of the instrument raises kinetic energies of parent (Ps+) and fragment ions (F+) to a level that the energy difference between parent and smallest fragment does not exceed 30% [32]. This might dramatically increase relative intensities of originally weak signals.

Among the identified FFAs, only two structural classes, earlier reported in rhizobia, namely saturated and unsaturated FAs, were detected. No hydroxylated and cyclic acids were detected. However, in general, the observed patterns of FFAs were in agreement with the data obtained with GC-MS. Thus, Theberge et al. also identified 16:0, 18:0 and 18:1 acids, although did not report 16:1 and 18:2 acids, described here [33]. A similar pattern was reported by Panday et al. in *Rhizobium pusense* [34], and, in general, confirmed here. It is important to note, that both authors report 19:0 acids (either aliphatic or cyclic), although these analytes could not be detected here. In a specialized Rhizobium species *R. selenireduscens*, Hunter et al. reported 3-hydroxystearic acid [35], which also was not found here. In the cells of *R. pseudoryzae*, Zhang et al. reported 18:1 and 19:0 cyclic acid, as well as methylated and hydroxylated species [36]. Based on the results obtained here, 18:1 cyclic acids could be potentially present in the cells analyzed in our study. Indeed, they are isomeric to detected 18:2 acid. However, as no separation was used, resolution of isomers is impossible. 

One of the reasons for the absence of methylated and hydroxylated FFAs in the list of annotated species in our study can be their low abundance, especially in their free form [7]; methylated FA may evaporate under MALDI conditions due to their higher volatility. Indeed, such acids are, for example, structural elements of Nod factors [37], which are rather low abundant molecules. Thus, to access these acids, Nod factors need to be isolated and hydrolyzed first [10].

Another possible reason for the absence of the above-mentioned substituted FFAs could have been a generally lower sensitivity of our method in comparison to standard GC-MS-based procedures. To address this issue, the extracts obtained by the same extraction procedures and from the same amounts of cell culture were analyzed by GC-MS after well-established derivatization procedures, i.e. methylation [6] and trimethylsilylation [38] protocols. Although no methyl ester-related signals could be detected in chromatograms, targeted search for trimethylsilyl (TMS) esters revealed only five FAs among those detected by MALDI-TOF-MS (Appendix A). However, the low intensities of these signals did not allow for quantitative estimation. Thus, for the system under investigation our method appeared essentially more sensitive in comparison to the conventional GC-MS-based approach. Indeed, due to selective ionization of barium monocarboxylates and omission of a derivatization step, detection of FAs under these conditions considerably improved in comparison. Evaluation of the GC-MS total ion current chromatograms (TICs) supports this conclusion. Indeed, the signals of glycerol, monopalmitoylglycerol and monostearylglycerol—compounds formed due to the activities of type 1 and 2 lipases [39]—dominated the TICs (Appendix A-3–S1-6), whereas these compounds were not detectable by MALDI-TOF-MS analysis of monolayers. Consequently, one option of the observed different selectivity of the analytical method could be competitive evaporation prior GC separation vs. selective ionization for MALDI-MS quantification. In our work, the anticipated sensitive detection of free intracellular FAs was targeted. This task, however, seems to be hardly possible by direct GC-MS: most of the described studies either target FFAs in cultural medium [40] or employ a saponification step, to address lipid-bound FAs [6]. Hence, employment of our protocol essentially extends the potential of mass spectrometry in FA analysis, and gives access to very sensitive, direct quantification of FFAs in rhizobial cells, as an improvement to earlier approaches. A partial loss of chain hydrophobicity, accompanying FA hydroxylation might negatively affect formation of monocarboxylate monolayers. To address this issue in more detail, test experiments with hydroxylated FA standards should be carried out in future work. 

### 4.2. Relative Quantification of FFAs in the R. Leguminosarum bv. Viciae RCAM1026 and S. Meliloti RCAM1021 Strains

The homogeneity of the groups and the differences between two species were assessed by multivariate statistics. Thereby, our quantification strategy was based on high linear dynamic ranges of analytes and low (<5%) interference of monoisotopic signals with isotopomers of other FAs [25]. To enable useful comparisons, raw data usually require normalization. In general, signal intensity depends on the relative amounts of the corresponding monocarboxylates at the specific point of a laser shot. In turn, due to different energy distribution patterns, the amounts of the ionizable salt determines the conformation of the monolayer—if it is straight or folded. As the monolayers obtained by the Langmuir technology, contain multiple types of FA anions, normalization to the major components of the FA patterns might give access to an appropriate correction factor for the variation of the monolayer conformation between spots on the MALDI target. Therefore, based on our previous experience [25], we probed the signals at *m/z* 393, 419 and 421, corresponding to palmitic, oleic and stearic acids, as well as the total ion current (TIC), for signal intensity normalization. In parallel, we compared two extraction protocols in respect of intra-group dispersion and confidence of the obtained results.

This comparison revealed *m/z* 421 as the optimal reference signal for normalization within the context of intra-group variance (Figure 5). Indeed, the numbers of differentially abundant (*p* ≤ 0.05) FFAs depended on the normalization strategy employed. Thus, 3 and 7 FFAs appeared significantly different for hexane- and buffer-based lysis, respectively, when the peak intensities were normalized to the *m/z* 393. However, these values decreased to 1 and 5, respectively, when normalization relied on *m/z* 419, and no differentially abundant features were found when total ion current (TIC) normalization was applied.

Surprisingly, despite lysis with hexane yielded lower signal intensities (Figure 3), it still resulted in a better separation of two rhizobial species in principal component analysis (PCA, Figure 5A,C) and less variance between biological replicates, as can be illustrated by corresponding heat maps (Figure 5B,D). Thereby, lysis with hexane yielded higher FA signal intensities in *R. leguminosarum* in comparison to *S. meliloti*, whereas lysis under aqueous conditions resulted in an inverse tendency indicating inter-species extraction efficiencies of FFAs. Hierarchical clustering of all individual spectra including all technical replicates, clearly indicated lysis with hexane as a preferable method with respect to precision and reproducibility (Figure 6). The other normalization strategies (summarized in the Appendix A) gave poorer separation between the compared groups, although the results were in agreement with the strategy employing normalization to *m/z* 421. Thereby, the differences between two lysis strategies were similar independently from normalization approach supporting the potential presence of a species-specific extraction effect (Appendix A–11). That was in agreement with distribution of non-normalized data (Appendix A). In agreement with this observation, normalization to *m/z* 421 resulted in statistically significant differences in the contents of FFAs, detected in *R. leguminosarum* and *S. meliloti* lysates. Thereby, six and five differentially abundant FFAs were observed when lysis was done with hexane (Appendix A) and water (Appendix A), respectively.

## 5. Conclusions

The protocol proposed here represents an essential advantage in analysis of free fatty acids. It is a superior tool for sensitive, precise and high-throughput screening of multiple samples, bacterial species, lines, mutants, ecotypes or specific ecological responses. Similar to the conventional GC-MS-based approach, our protocol enables the analysis of FFAs with high sensitivity and precision, which we attribute to the high regularity of monocarboxylate monolayers formed directly on the MALDI target. Moreover, application of Langmuir technology in combination with MALDI-TOF-MS allows large batch sizes, which are basically restricted only with the capacity of employed MALDI targets, typically 384 analyses including calibration samples. Such a high throughput is hardly accessible by GC-MS, which is obviously less favored in large-scale screening experiments. One needs to keep in mind, however, that the absence of chromatographic separation and hence, the inability to separate isomeric acids is a principal, intrinsic bottleneck of MALDI-TOF-MS. Thus, complementation of our approach with other methods, giving access to the isomer composition of the samples and selected based on preliminary screening, is advantageous.

## Figures and Tables

**Figure 1 mps-03-00036-f001:**
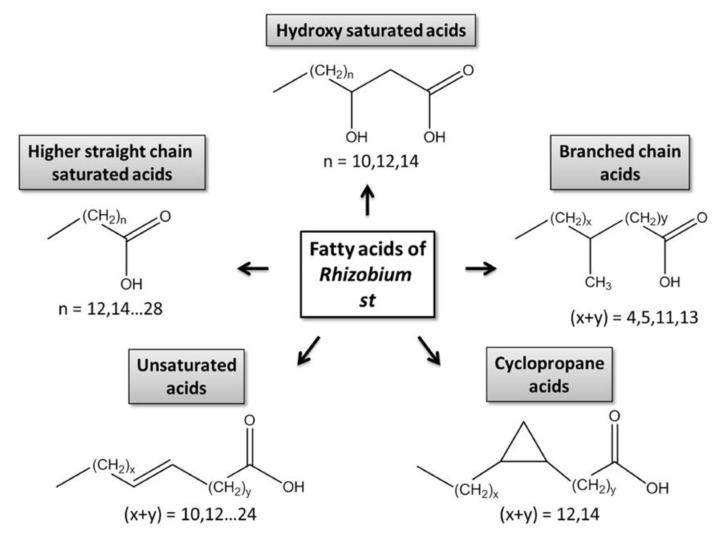
Structural classes of bacterial fatty acids.

**Figure 2 mps-03-00036-f002:**
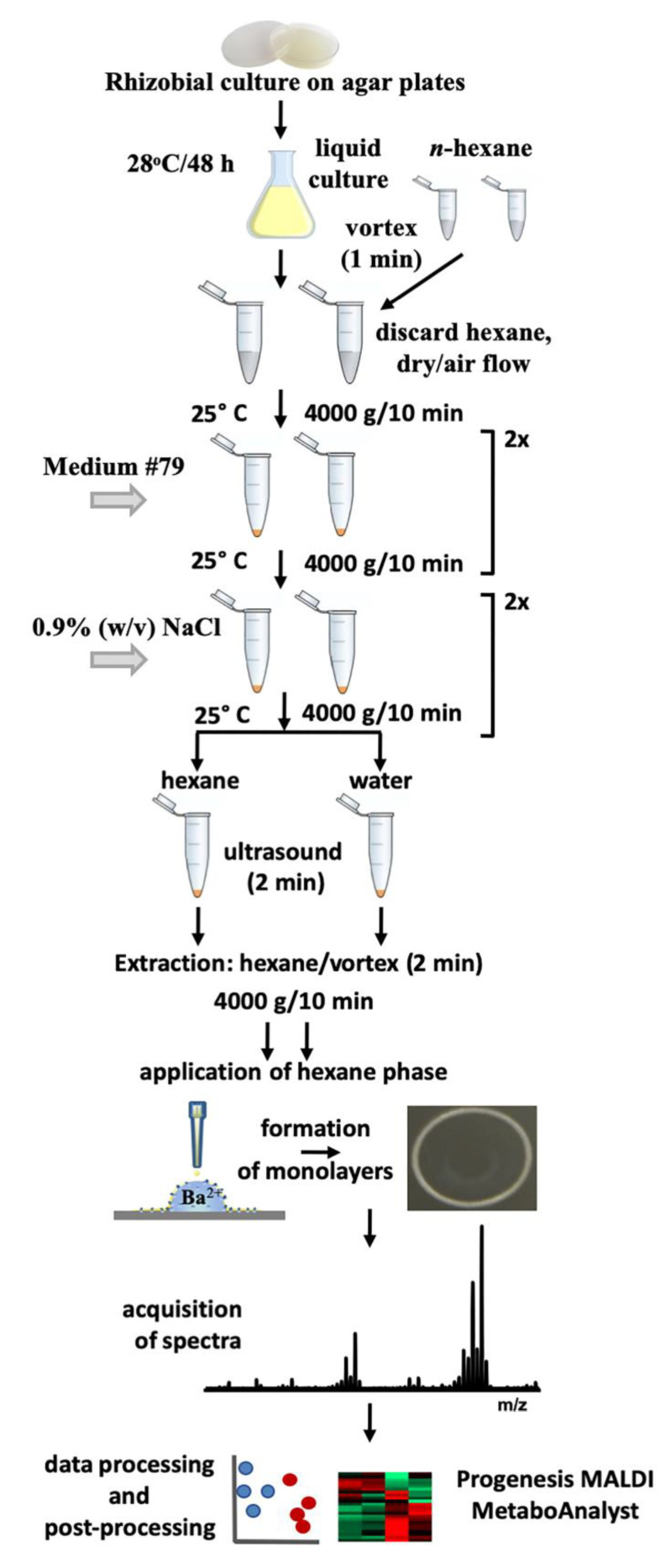
Experimental setup, applied for analysis of rhizobial free fatty acids by the combination of Langmuir technology and matrix-assisted laser desorption/ionization-time of flight mass spectrometry (MALDI-TOF-MS).

**Figure 3 mps-03-00036-f003:**
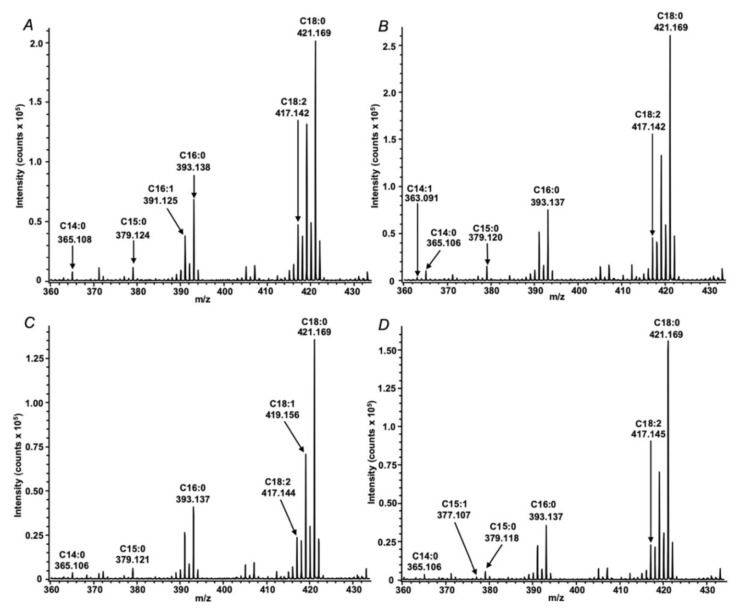
MALDI-TOF mass spectra of extracts obtained from aqueous (**A**,**B**) and *n*-hexane (**C**,**D**) lysates of *Rhizobium leguminosarum* bv. viciae RCAM1026 (**A**,**C**) and *Sinorhizobium meliloti* RCAM1021 (**B**,**D**) cells.

**Figure 4 mps-03-00036-f004:**
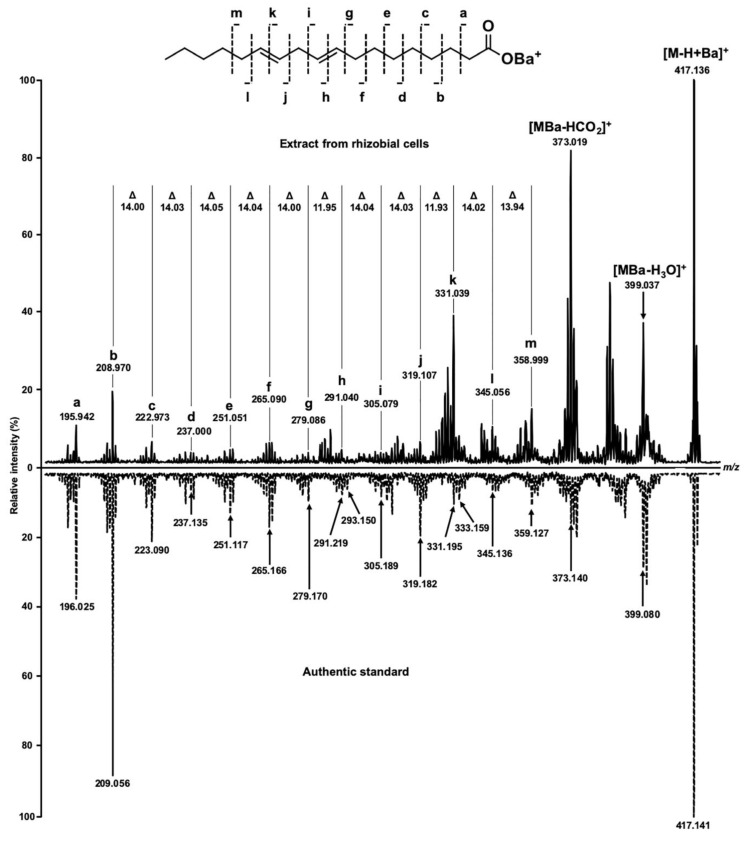
MS/MS spectra of *m/z* 417.14, corresponding to [M-H+Ba]^+^ ion of linoleic acid. The authentic standard of linoleic acid was dissolved in *n*-hexane and applied on MALDI target in the same way as the extract from rhizobial cells. The double bonds were localized by characteristic methine mass shifts (12 *m/z*), which could be annotated to the fragments h and k.

**Figure 5 mps-03-00036-f005:**
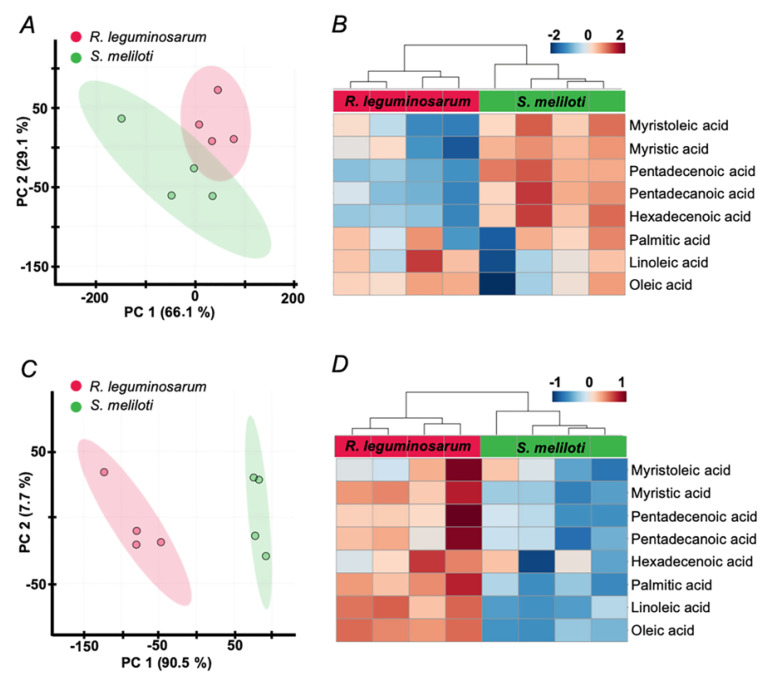
Principal component analysis (PCA) score plots (**A**,**C**), built for the first two principal components and heat maps (**B**,**D**) for the patterns of FFAs content alterations across *R. leguminosarum* and *S. meliloti* samples, obtained by extraction from the aqueous (**A**,**B**) and *n*-hexane lysates (**C**,**D**). The data were processed in the Progenesis MALDI software with data normalization on the intensities (peak heights) of the signal at *m/z* 421.17 within a given mass range. The data matrix was further analyzed by the MetaboAnalyst 4.0 on-line software tools (https://www.metaboanalyst.ca/). Individual columns in heat maps represent biological replicates.

**Figure 6 mps-03-00036-f006:**
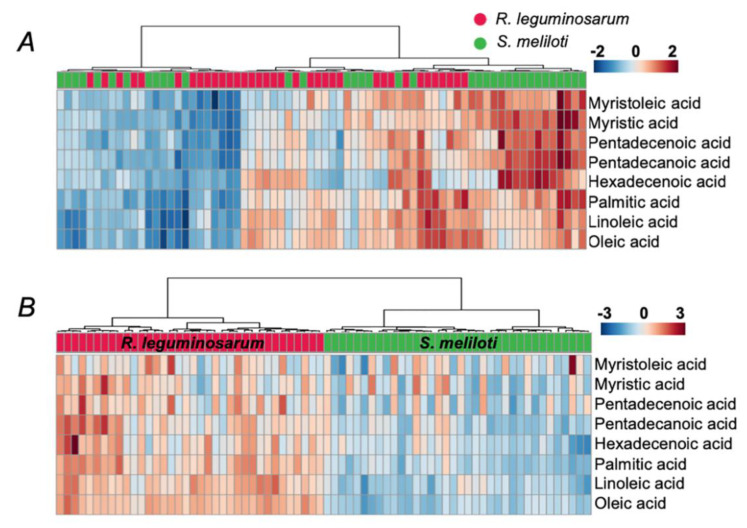
Heat maps of FFA patterns across all spectra (biological, spotting and acquisition replicates), acquired for *R. leguminosarum* and *S. meliloti* samples, obtained by the extraction from aqueous lysates (**A**) and *n*-hexane lysates (**B**). Data were processed in the Progenesis MALDI software with data normalization to the intensities (peak heights) of the signal at m/z 421.17 within a given mass range. The data matrix was further analyzed by the MetaboAnalyst 4.0 on-line software platform (https://www.metaboanalyst.ca/). Technical replicates (showed as individual columns) demonstrate a higher degree of similarity in comparison to the biological ones.

**Table 1 mps-03-00036-t001:** Annotation of free fatty acids (FFAs) in *n*-hexane extracts of aqueous and hexane lysates of *Rhizobium leguminosarum* bv. viciae RCAM1026 and *Sinorhizobium meliloti* RCAM1021 cells.

#	*m/z* [M-H+Ba]^+^	Elemental Composition	Annotation of Analytes					Samples	
			Annotation (Label)	Error (ppm)	Isotopic Patterns	MS^2^	Tentative Identification	RL	SM
**1**	363.091	C_14_H_25_O_2_Ba^+^	C14:1	8.3	+	-	Myristoleic acid	+	+
**2**	365.106	C_14_H_27_O_2_Ba^+^	C14:0	5.5	+	+	Myristic acid	+	+
**3**	377.106	C_15_H_27_O_2_Ba^+^	C15:1	2.7	+	-	Pentadecenoic acid	+	+
**4**	379.122	C_15_H_29_O_2_Ba^+^	C15:0	−5.3	+	+	Pentadecanoic acid	+	+
**5**	391.122	C_16_H_29_O_2_Ba^+^	C16:1	5.1	+	+	Hexadecenoic acid	+	+
**6**	393.138	C_16_H_31_O_2_Ba^+^	C16:0	−2.6	+	+	Palmitic acid	+	+
**7**	417.138	C_18_H_31_O_2_Ba^+^	C18:2	9.6	+	+	Linoleic acid	+	+
**8**	419.153	C_18_H_33_O_2_Ba^+^	C18:1	7.2	+	+	Oleic acid	+	+
**9**	421.169	C_18_H_35_O_2_Ba^+^	C18:0	2.4	+	+	Stearic acid	+	+

RL, Rhizobium leguminosarum bv. viciae RCAM1026; SM, Sinorhizobium meliloti RCAM1021.

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
