# Peer review of "High-Throughput Fingerprinting of Rhizobial Free Fatty Acids by Chemical Thin-Film Deposition and Matrix-Assisted Laser Desorption/Ionization Mass Spectrometry"

_mps, 2020, doi:10.3390/mps3020036_

Round 1
Reviewer 1 Report
The manuscript describes that a method for the analysis of rhizobial free fatty acids using a thin film chemical deposition combined with MALDI-MS.
It is very informative. After the following corrections, it will be ready for publication.
It would be better if the authors comment about the reason that DHB was used as matrix in the current investigation.
line 55
of -> or
Line 360 For the description of Figure 4, it would be better if the authors comment on the cleavage of the double bonds?
Line 367
In Table 1, the theoretical monoisotopic m/z values need to be provided. In addition, the error (ppm) values need to be checked again.
Line 353
The authors mentioned that the lysis under aqueous conditions yielded better spectral intensitiesa and better signal to noise ratios than those from hexane lysis. The intensities are higher from aqueous conditions. However, S/N ratios look the same. It would be better if the authors provide the actural S/N values.
Line 409
It would be better if the authors mention about the results using the other peaks other than the peak at m/z 421.
Author Response
We thank the reviewer for the thoughtful review and highly appreciate the valuable comments and suggestions to improve the manuscript. Following these advices we performed all required changes in corresponding sections, as indicated in the following rebuttal addressing each aspect.
Reviewer: 1
Remarks
Remark 1:“It would be better if the authors comment about the reason that DHB was used as matrix in the current investigation”
Answer:DHB was supplemented to the aqueous phase in order to improve energy distribution and, thereby, ionization efficiency. DHB is advantageous in comparison to other matrices due to its good solubility in water. The text is changed accordingly (lines 119 -123):
“Ionization efficiency can be further increased by supplementation of the aqueous phase with conventional MALDI matrices, based on substituted benzoic or cinnamic acids, which improve energy distribution in the sample upon the laser shot [28]. In this context, 2,5-dihydroxybenzoic acid (DHB) proved to be the best choice due to its good solubility in water [25].”
Remark 2:“line 55. of -> or”
Answer:Changed accordingly.
Remark 3:“Line 360. For the description of Figure 4, it would be better if the authors comment on the cleavage of the double bonds?”
Answer:Yes, we agree with the reviewer – it will be better to provide this information.The mass increment between the fragments h and g, as well as k and j, is 12 mass units. This corresponds to methine group, indicating the presence of a double bond. The following text is added to the legend of Figure 4:
“The double bonds were localized by characteristic methine mass shifts (12 m/z), which could be annotated to the fragments h and k” (lines 450 - 451).
Remark 4:“Line 367. In Table 1, the theoretical monoisotopic m/z values need to be provided. In addition, the error (ppm) values need to be checked again.”
Answer:Corrected and checked accordingly.
Remark5:“Line 353. The authors mentioned that the lysis under aqueous conditions yielded better spectral intensities and better signal to noise ratios than those from hexane lysis. The intensities are higher from aqueous conditions. However, S/N ratios look the same. It would be better if the authors provide the actual S/N values.”
Answer:The values are provided now in the Supplementary table S1-2. The corresponding reference is given in text (line 365).
Remark6:“Line 409. It would be better if the authors mention about the results using the other peaks other than the peak at m/z 421.”
Answer:We agree with the reviewer, this information needs to be provided, and we do it. The following text was appended to the text:
“Indeed, the numbers of differentially abundant (p ≤ 0.05) FFAs depended on the normalization strategy employed. Thus, 3 and 7 FFAs appeared significantly different for hexane- and buffer-based lysis, respectively, when the peak intensities were normalized to the m/z 393. However, these values decreased to 1 and 5, respectively, when normalization relied on m/z 419, and no differentially abundant features were found when total ion current (TIC) normalization was applied.” (lines 453 - 458).

Reviewer 2 Report
The manuscript gives a detailled description of a method previously develop for fingerprinting free fatty acids. This method combining Langmir technology with MALDI TOF-MS measurements was applied to the comparison of FFA content of two rhizobia species.
The paper is well written and even if some interpretations of the results need to be reviewed (MS/MS, conclusion non supported by the data) and some data missing (MS settings), the implementation of the method seems to be effective.
However, there is a high lack of originality/novelty compared to what the authors published in Analytical Chemistry journal in 2019. Their previous paper already described a large range of application of this method (especially in the supporting information). The application of this methodology to two other species using two different extraction solvents is not sufficient novelty, it is more an application of the method published elsewhere than a new method.
Author Response
We thank the reviewer for the thoughtful review and highly appreciate the valuable comments and suggestions to improve the manuscript. Following these advices we performed all required changes in corresponding sections, as indicated in the following rebuttal addressing each aspect.
Reviewer: 2
Remarks
Remark 1:“The paper is well written and even if some interpretations of the results need to be reviewed (MS/MS, conclusion non supported by the data) and some data missing (MS settings), the implementation of the method seems to be effective.”
Answer: We agree with the reviewer – the interpretation of a FA fragmentation pattern (exemplified by the tandem mass spectrum of the [M-H+Ba]+ ion representing linoleic acid in Figure 4) needs to be provided. Now it is given in the legend for Figure 4:
“The double bonds were localized by characteristic methine mass shifts (12 m/z), which could be annotated to the fragments h and k” (lines 450-451).
Additionally, we provide the whole information on MS method settings (now it is Table S1-1 in the Supplementary information 1). The corresponding reference is provided in text (lines 294-295).
Remark 2:“However, there is a high lack of originality/novelty compared to what the authors published in Analytical Chemistry journal in 2019. Their previous paper already described a large range of application of this method (especially in the supporting information). The application of this methodology to two other species using two different extraction solvents is not sufficient novelty, it is more an application of the method published elsewhere than a new method.”
Answer: The reviewer is right, we need to better verbalize the scope of the study and show our focus. Indeed, we reported the methodology behind the proposed protocol in Analytical chemistry last year. And, of course, to be aware, that we were not dealing with artifacts, we verified applicability of this analytical solution to a broad range of biological matrices. We are convinced that the potential of the Langmuir technology in modern analytical chemistry was comprehensively good shown in that paper. But, we did not give there any protocol-specific data and did not illustrate quantitative application, because it was behind the scope of that paper. Here we are proposing a protocol, relying at the analytical step on the Langmuir/MALDI technology, and comprising a number of other steps. And here, our aim was to show the step-by-step implementation of the published concept in the real method. Here, for the first time we propose the quantitative comparison, based on the data, acquired by the Langmuir/MALDI technology, i.e. for the first time in the form of a protocol, showing how it is working for answering biological questions. In this sense, we find our manuscript novel enough, it is definitely not re-writing of the previous paper for new material – this manuscript has another scope. It is pretty good said in the abstract, but we changed/extended the last sentence in the introductory part to make the statement more clear and focused:
“Therefore, here we implement this approach in a newly developed protocol for the analysis of FFA composition in rhizobial bacteria and propose a comprehensive quantitative solution for fingerprinting of FFAs in cultured cells by MALDI-TOF-MS. Thereby, we demonstrate for the first time the applicability of the Langmuir/MALDI-TOF-MS approach to answering real biological questions.”(lines 92 - 96).

Reviewer 3 Report
Gladchuk et al. here proposed a protocol for the high-throughput relative quantification of bacterial free fatty acids by the employment of MALDI-TOF MS applied on samples prepared harnessing the Langmuir technique. All the study has been done in the perspective of the detection and the analysis of free fatty acids in a simple and straightforward manner replacing some currently used procedures involving, among others, GC-MS, HILIC, RP-HILIC applied to ESI-MS, etc... In order to verify the feasibility and the analytical potential of their approach, the authors focused on two plant associated bacteria, i.e. Rhizobium leguminosarum and Sinorhizobium meliloti.
This reviewer strongly deems that the overall work is well performed and described, nevertheless some major points should be addressed prior publication as some important points are missing or unclear.
The main limit of the described procedure seems to be the undetectability of non-hydroxylated and cyclic acids. The authors gave a sort of explanation taking in account the low abundance of such fatty acids in their free form. Theoretically this is not true for all bacteria, as it is possible to find huge amount of free hydroxylated fatty acids coming from free lipopolysaccharide glycolipid moiety or molecules derived from bacterial quorum sensing. The authors should take in account this point as it is surprising to not detect hydroxylated fatty acids which can be, instead, easily detected by GC-MS analysis.
In their analyses, the authors found the typical fatty acids of Rhizobia genus, namely long-chained and medium long-chained fatty acids, which actually could have been easily and fast detected via GC-MS after proper derivatization (such as to methyl esters) followed by hexane extraction. The derivatization in methyl esters, as example, furnishes also information on the hydroxylated acyl chains that were not detected by the authors. Therefore, I strongly invite the authors to investigate this aspect and compare the results with those obtained by their protocol, and implement the information in their manuscript.
Contextually, the authors state that this methodology can potentially help to appreciate also short-chained fatty acids, but actually they analyzed two Rhizobia strains known to possess only medium and long-chained fatty acids. It is mandatory to demonstrate the real potential of this methodology in allowing the detection of all free fatty acids. To do this, the employment of standards or even of bacteria known to produce short-chained free fatty acids is necessary prior publication.
Figure 4. The authors should give an explanation for the evident peaks at around m/z 300 (which is quite strange) and 325.
Figure 2. This figure does not properly match with what is described in the Procedure paragraph. As example, it is not clear how and why the hexane is used before the application of the rhizobial cultures to the tubes.
Have the authors considered the possibility to lose part of the free fatty acids (especially the hydroxylated ones) in their pre-cleaning step? Please comment on this.
Table 1. Please indicate in the legend the entire name of the acronym RL and SM
Line 50. Should be “accessed either directly (FFAs) or after degradation”
Author Response
We thank the reviewer for the thoughtful review and highly appreciate the valuable comments and suggestions to improve the manuscript. Following these advices we performed all required changes in corresponding sections, as indicated in the following rebuttal addressing each aspect.
Reviewer: 3
Remarks
Remark 1:“In their analyses, the authors found the typical fatty acids of Rhizobia genus, namely long-chained and medium long-chained fatty acids, which actually could have been easily and fast detected via GC-MS after proper derivatization (such as to methyl esters) followed by hexane extraction. The derivatization in methyl esters, as example, furnishes also information on the hydroxylated acyl chains that were not detected by the authors. Therefore, I strongly invite the authors to investigate this aspect and compare the results with those obtained by their protocol, and implement the information in their manuscript.”
Answer: We absolutely agree with the reviewer –this comparison might be a good way to cross-validate our results. Moreover, such a study might give valuable information on selectivity and sensitivity of our MALDI-TOF-MS-based method. It is especially important in the context of the object (and here we also agree with the reviewer): by this protocol we analyze not medium, but cells, which are not featured with high equilibrium levels of FFAs. This specific feature in comparison to plants and animals was not addressed in our previous work. Therefore, it is time to do it. After a thorough exploration of literature, we failed to find analysis of free acids in cell by GC-MS without any enrichment, fractionation etc. All such analysis relay either on lipid hydrolysis prior to derivatization, or profiling of FFAs in culture medium. Expectedly, our method was clearly superior in comparison to GC-MS both in the sense of selectivity and sensitivity. Obviously, the samples amounts used do not allow identification of hydroxylated FFAs. All necessary corrections in text are done:
“Another possible reason for the absence of the above mentioned substituted FFAs could be generally lower sensitivity of our method in comparison to standard GC-MS-based procedures. To address this issue, the extracts obtained by the same procedures and from the same amounts of cell culture were analyzed by GC-MS after appropriate derivatization according well-established methylation [6] and trimethylsylilation [34] protocols. Although no methyl ester-related signals could be detected in chromatograms, targeted search for trimethylsilyl (TMS) esters revealed only five FAs among those detected by MALDI-TOF-MS (Supplementary information 1, Tables S1-3 – S1-6). However, the intensities of these signals were low and did not allow any quantitative estimation. Thus, for the system under investigation our method turned to be essentially more sensitive in comparison to the conventional GC-MS-based approach. Indeed, due to selective ionization of barium monocarboxylates and omission of a derivatization step, detection of FAs under these conditions considerably improved in comparison. Evaluation of the GC-MS total ion current chromatograms (TICs) supports this conclusion. Indeed, the signals of glycerol, monopalmitoyl glycerol and monostearyl glycerol – compounds formed due to the activities of type 1 and 2 lipases [35]-dominated the TICs (Tables S1-3 – S1-6), whereas these compounds were not detectable by MALDI-TOF-MS analysis of monolayers. Consequently, one option of the observed different selectivity of the analytical method could be competitive evaporation prior GC separation vs. selective ionization prior MALDI-MS quantification. In our work, sensitive detection of free intracellular FAs were targeted. This task, however, seems to be hardly solvable by direct GC-MS: the most of the described studies either target FFAs in cultural medium [36] or employ a saponification step, to address lipid-bound FAs [6]. Hence, employment of our protocol essentially extends the potential of mass spectrometry in FA analysis, and gives access to very sensitive, direct quantification of FFAs in rhizobial cells, as an improvement to earlier approaches. A partial loss of chain hydrophobicity, accompanying FA hydroxylation might negatively affect formation of monocarboxylate monolayers. To address this issue in more detail, test experiments with hydroxylated FA standards should be carried out in future work.”(lines 406 - 431).
Remark 2:“Contextually, the authors state that this methodology can potentially help to appreciate also short-chained fatty acids, but actually they analyzed two Rhizobia strains known to possess only medium and long-chained fatty acids. It is mandatory to demonstrate the real potential of this methodology in allowing the detection of all free fatty acids. To do this, the employment of standards or even of bacteria known to produce short-chained free fatty acids is necessary prior publication.”
Answer: We agree with the reviewer, this information needs tobe provided. We performed additional experiments with a mixture of C6:0, C8:0, C10:0 and C12:0 acids. The experiments revealed detection of the acids with carbon chains of 10 C-atoms or longer. The following statement is added to the text:
“In additional experiments we addressed the limitations of the method in the context of the acid chain length. We found, that the C6:0 and C8:0 acids are not detectable by this protocol, whereas the signal of C10:0 demonstrates lower intensity in comparison to higher homologs (C12:0 and higher).” (lines 136 - 143).
Remark 3:“Figure 4. The authors should give an explanation for the evident peaks at around m/z 300 (which is quite strange) and 325.”
Answer: The signal at m/z 325 represent DHB matrix that was confirmed by tandem mass spectrometric experiments. The m/z 416.97 was present in the mass spectra of blanks (hexane instead of hexane extract) and was selected by the TIS device of the mass spectrometer. These experiments, including also additional controls, missing individual steps of the procedure, are summarized now in Table S1-7. A broad signal at m/z 300 was absent in controls, but present in the majority of acquired MS/MS spectra of [M-H+Ba]+ ions. Based on its form, it is difficult to attribute it to any specific mass. Thechangesinthetextaredone:
“In additional experiments some relatively abundant signals, like m/z 325 could be attributed to DHB matrix (Table S1-7, Figure S1-7). However, the origin of the broad signal at m/z 300 could not be understood, although it was present in the most of analyzed MS/MS spectra.” (lines 377-380).
Remark 4:“Figure 2. This figure does not properly match with what is described in the Procedure paragraph. As example, it is not clear how and why the hexane is used before the application of the rhizobial cultures to the tubes.”
Answer:We agree with the reviewer, and add the missing step to Figure 2. The wash of tubes with hexane prior to extraction is necessary to remove potential hexane-soluble contaminations. The corresponding sentence is added to the text:
“This step is necessary to wash out potential hexane-soluble contaminations, present in tubes.” (line 239-240).
Remark 5: “Have the authors considered the possibility to lose part of the free fatty acids (especially the hydroxylated ones) in their pre-cleaning step? Please comment on this.”
Answer: Of course, a partial loss of chain hydrophobicity, accompanying FA hydroxylation might negatively affect formation of monocarboxylate monolayer. To address this issue in more details test experiments with standard hydroxylated FAs can be performed. This was, however, behind the scope of this work, as the main aim of the protocol – to gives access to fast, sensitive and specific analysis of rhizobial FFAs, is achieved. The appropriate changes in text are done:
“A partial loss of chain hydrophobicity, accompanying FA hydroxylation might negatively affect formation of monocarboxylate monolayers. To address this issue in more detail, test experiments with hydroxylated FA standards should be carried out in future work.” (lines 428 - 431).
Remark 6:“Table 1. Please indicate in the legend the entire name of the acronym RL and SM”
Answer: The entire names are indicated in the legend below the table (line 383).
Remark 7: “Line 50. Should be “accessed either directly (FFAs) or after degradation””
Answer: Corrected accordingly.

Round 2
Reviewer 2 Report
See remarks in red color in the attached file

Author Response
We thank the reviewer for the thoughtful review and highly appreciate the valuable comments and suggestions to improve the manuscript. Following these advices we performed all required changes in corresponding sections, as indicated in the following rebuttal addressing each aspect.
Remarks
Remark 1:“From my experience in MS/MS analyses of FA ions, I suspect some wrong assignments of fragment ions from the MS/MS experiment of m/z 417:
- from [FA-H+Ba]+ ions, it is not possible to loose CO2 or H2O without losing the Ba2+ ion.
Response: We assume, that cleavage of H2O from a [FA-H+Ba]+ ion without dissociation of Ba2+ can be underlied by a keto-enol tautomeric transition, which is accompanied by transfer of a hydrogen atom from the α-methylene group to the carboxyl oxygen, with subsequent dehydration and formation of barium oxyacetylenide:
The following text is added: “The second signal might be interpreted as a water loss, underlied by a keto-enol tautomeric transition, which is accompanied by transfer of a hydrogen atom from the α-methylene group to the carboxyl oxygen, with subsequent dehydration and formation of barium oxyacetylenide” (lines 382-385).
Remark 2:“The fragment ions at m/z 399 and 373 seems to come from DHB ion at m/z=417 (selected at the same time – see your Table 1-7).
Response: Weunderstand the doubt of the reviewer – chemically it might look not so straightforward. That is why, for such statement we needed, of course, to rely on some previous data. Indeed, the loss of CO2 from the ion pair without its disturbance is pretty good described tor FAs. Thus, Afonsoetal (J Mass Spectrom. 2005; 40(3): 342–349) reported a signal at m/z 300 in thetandem mass spectrum of m/z 344, corresponding to the [(M-H)+CuII]+ ion of oleic acid. In their work, the authors gave a clear interpretation of this signal as originating from direct loss of CO2 without dissociation of CuII. Further, Crockettet al.(JAm Soc Mass Spectrom.1990, 1(2): 183-191) showed formation of [M-H+Ba]+ions in the CAD spectra of octadecadienic acids with different localization of a double bond. The authors interpret such a signal at m/z 373 as a direct loss of CO2 without dissociation of the Ba2+ ion. In both cases the authors proposed a mechanism, employing coordination of the metal by the double bond, formed after the CO2 loss. However, we observed the loss of 44 u in tandem mass spectra of saturated FAs. We annotated this signal as a CO2 loss. This proposal was accepted by three reviewers of the leading analytical journal – Analytical chemistry, invited by the editor – Prof. Renato Zenobi (see Fig. S1-6BinAnalChem. 2019. 91(2): 1636-1643).Therefore, keeping in mind the published data we propose the following mechanism of the loss of CO2 from barium monocarboxylates:
The following text is added: “The spectra dominated with intense signals at m/z 373.02 and 399.04. The first of them corresponds to the loss of CO2, formed by the mechanism, earlier reported by Afonso et al. [30] and Crockett et al. [31]” (lines 380-382).
Remark 3:“-the Lift system for MS/MS experiment on this MALDI TOF/TOF instrument dissociates the ion intensities (and S/N ratios) between the MS ion selection/acquisition step and the MS/MS acquisition step (according to the background differences, this is the case). This difference have the effect to increase “artificially” the ion intensities of normally very low abundant ions (especially from the matrix in that case).
Response: We thank reviewer for this remark. The correspondent remark is implemented in the text: “One needs to keep in mind, however, that the LIFT system of the instrument raises kinetic energies of parent (Ps+) and fragment ions (F+) to a level that the energy difference between parent and smallest fragment does not exceed 30% [32]. This might dramatically increase relative intensities of originally weak signals” (lines 389 – 392).
Remark 4:“-there is a high m/z error in the ion assignement: for example, [FA-H-H2O+Ba]+ ,if it exists, would be detected at m/z 399.127 (and not 398.85)
- during MS/MS of non-saturated FA, it is absolutely not possible to break a double bond.Consequently, fragment ions h or k cannot be detected.”
Response:We completely agree with the reviewer, this error is too big, we apologize about this. We re-acquired the spectra – Figure 4 is replaced now. Now, mass accuracy (also for the losses of CO2 and H2O) meets specifications of the instrument (besides one increment – l to m) and in agreement with our earlier published data (Anal Chem. 2019. 91(2): 1636-1643). This is obviously enough for identification with authentic standard.Importantly, in the context of the following remark, the ions h and k are reliably detected now.
Remark 5:“The position of double bonds is very critical by MS/MS, especially from cationized FA. Than I suggest that you do not discuss about MS/MS because your data do not support your discussion
Response:MS/MS-based structure characterization of fatty acids is perfectly described in literature. The reviewer is right, some authors note the absence of fragmentation at the position of a double bond (JAm Soc Mass Spectrom.2018, 29(8): 1688-1699; JAm Soc Mass Spectrom.1999, 10(7):600-612; JAm Soc Mass Spectrom.2008, 19(11): 1673-1680; Rapid Commun Mass Spectrom.2008, 22(13):2125-2133). On the other hand, others report characteristic signals, clearly seen in fragmentation patterns of FAs and interpretable in the context of fragmentation at the position of double bonds (JAm Soc Mass Spectrom.1990, 1(2): 183-191;Anal Chem.1987, 59(11): 1576-1582; JAm Soc Mass Spectrom.2014, 25: 1917-1926; JMass Spectrom.2017, 52(5):271-282). Thus, for example, Crockett et al (JAm Soc Mass Spectrom.1990, 1(2): 183-191) andAdams et al(Anal Chem.1987, 59(11): 1576-1582) report the presence of signals in the FAB spectra (i.e. acquired by the technology, to some extent comparable to the conditions of MALDI) of cation adducts of FAs, formed as the result of fragmentation at the double bond. On the other hand, Thomasetalobserved product ions resulted from cleavages at the sites of double bonds in theCIDspectrumof the Δ4,7,10,13,16,19 22:6[M-2H+Na]-ion(JAm Soc Mass Spectrom.2014, 25: 1917-1926).
In our particular case, we assume formation of the ions h and k not via the direct cleavage of the double bond, but as a result of a double bond migration to the neighboring carbon atom with subsequent σ-cleavage of a single bond formed. As this migration can occur in two directions in respect to the original double bond position, fragmentation yields two ions, seen in the spectra as two signals with a mass difference of 2 u (Scheme below). Thus, in the tandem mass spectra of [M-H+Ba]+ions of an unsaturated FA two signals with mass differences of 12 and 14 u in comparison to the proximal σ-cleavages can be expected. In context of these considerations, in the MS/MS spectra of linoleic acid we expect couples of signals at the m/z 291/293 and 331/333 for the double bonds at С9-С10 and С12-С13, respectively. This expectation is supported by the observation of the m/z 333.1 in a pseudo-MS3 spectrum of [C18:2+Ba-H]+(Anal Bioanal Chem. 2018; 410(5): 1435–1444). The authors interpreted this signal in the context of double bond migration. Here we observe the both pairs of ions, that allows unambiguous identification of the double bond position. Exactly this strategy we employed in our previous work. It was successfully reviewed by Analytical chemistry, that is why here we refer to this work without further explanations in text.
The reference to the mechanism is added: “as described in our previous work [25] based on the mechanism assuming double bond migration with subsequent σ-cleavage [29]” (lines 375-377).
Remark 6:“OK, I understand that you want to go deeper in the development of this analytical protocol. To be whole, you need to discuss all parts. And for that, you should discuss about the isotopic distribution of Ba that influences the ion intensity you detected for all peaks on your MS spectra (see MS spectrum simulated for Ba+). The MS system you used is not resolutive enough to distinguish one MS peak of isotopic contribution to those from another compound with one or more double bonds (mass accuracy is not sufficient). You have to take care about that. Also, it may bring another point of view to explain why one normalization could work or not for data analyses.”
Answer:This remark of the reviewer is clear for us, as we responded to such a question when publishing our previous paper (Anal Chem. 2019. 91(2): 1636-1643). There, in a model experiment, we have shown, that notmorethan 5% ofinterferencecould be seen betweenisotopicpatternsofstearicand oleic acid.In that work, when the mixtures 16:0/16:1 and 18:0/18:1 were analyzed, we have seen, that ingeneral, theratiosofmonoisotopicpeakintensitiesgenerallycorrespondtotheratiosofFAsinasample, andbothgroupsofanalytesdemonstrated linearresponse (R2 ≥ 0.97) forpotentiallyaffectedacids. In the context of the reviewer’s remark, we add the corresponding reference in text: “Thereby, our quantification strategy was based on high linear dynamic ranges of analytes and low (< 5%) interference of monoisotopic signals with isotopomers of other FAs [25].” (lines446 – 448).

Reviewer 3 Report
In their revised manuscript, the authors have addressed my concerns. I therefore support publishing the manuscript in its current form.Author Response
We thank the reviewer for his time and critical remarks